# Corneal endothelial cell density loss following glaucoma surgery alone or in combination with cataract surgery: a systematic review protocol

Clarissa E H Fang,[1] Peng Tee Khaw,[1,2] Rashmi G Mathew,[1,3] Christin Henein ![ORCID][1,2]

¹University College London Institute of Ophthalmology, London, UK
²NIHR Moorfields Biomedical Research Centre, London, UK
³Moorfields Eye Hospital NHS Foundation Trust, London, UK

**Correspondence to**
Dr Christin Henein;
c.henein@ucl.ac.uk

## ABSTRACT

**Objective** We aim to systematically assess and compare corneal endothelial cell density (ECD) loss in patients with glaucoma following glaucoma surgery and cataract surgery.

**Introduction** Corneal ECD loss may occur due to intraoperative surgical trauma in glaucoma surgery or postoperatively with chronic endothelial cell trauma or irritation. Corneal oedema and decompensation after aqueous shunt glaucoma surgery has been reported but the long-term ECD loss is still unknown.

**Inclusion criteria** Trabeculectomy, glaucoma filtration surgery or microinvasive glaucoma surgery in adults with ocular hypertension, primary and secondary open angle glaucoma, normal tension glaucoma and angle-closure glaucoma. Participants with pre-existing corneal disease will be excluded. Glaucoma laser treatments and peripheral iridotomy will be excluded. The outcomes include preoperative and postoperative corneal ECD, percentage change of corneal ECD and adverse events.

**Methods** We will conduct an electronic database search for randomised controlled trials, prospective non-randomised studies, observational studies in MEDLINE, EMBASE, Cochrane Central Register of Controlled Trials (CENTRAL), ClinicalTrials.gov and The International Prospective Register of Systematic Reviews (PROSPERO). Eligibility criteria will include quantitative articles published after and including the year 2000, written in English and containing data on ECD loss. Two independent reviewers will screen titles and abstracts and extract data from full texts, reporting outcomes according to Preferred Reporting Items for Systematic Reviews and Meta-Analyses (PRISMA) guidelines. Data extraction of key characteristics will be completed using customised forms. Methodological quality will be assessed using the Joanna Briggs Institute critical appraisal forms.

**Ethics and dissemination** Ethics approval is not required for this review, as it will only include published data. Findings will be published in a peer-reviewed journal and disseminated across ophthalmic networks.

**PROSPERO registration number** PROSPERO CRD42020192303.

## Strengths and limitations of this study

► This protocol adheres to the Preferred Reporting Items for Systematic Reviews and Meta-Analysis protocols and has been published prospectively in PROSPERO database.
► The review will comprehensively assess published peer-reviewed manuscripts assessing endothelial cell loss and adverse events following glaucoma surgery.
► A potential limitation might be the paucity of high-quality glaucoma surgery trials measuring endothelial cell density loss.
► Due to the expected heterogeneity in study methods, it is unlikely that a network meta-analysis will be possible.

surface that regulates stromal hydration. This squamous cell layer does not regenerate and if damaged, the cornea becomes oedematous, causing loss of corneal clarity thus compromising vision, negatively affecting quality of life. Physiologically, endothelial cell density (ECD) reduces with age at a rate 0.5% per year.[1] Glaucoma is a known condition that predisposes to endothelial cell damage, although the underlying mechanisms are not well elucidated.[2] Certain types of glaucoma are more predisposed to ECD loss such as when there is a sharp rise in intraocular pressure (IOP) in acute angle closure.[3] Glaucoma has also been associated with poor cornea graft survival.[4 5] High levels of IOP as well as long duration of high IOP are associated with more ECD loss.[2 6 7] It remains inconclusive whether glaucoma medications cause endothelial cell loss.[2]

Glaucoma surgery is effective in reducing IOP and preventing the progression of optic nerve damage and visual field loss. Trabeculectomy, or bleb filtration surgery, is considered the gold standard and is frequently performed for medically uncontrolled

## INTRODUCTION

The corneal endothelium is a monolayer of hexagonal cells lining the posterior corneal

glaucoma. Aqueous drainage shunts have conventionally been used in cases of uncontrolled glaucoma following unsuccessful trabeculectomy. However, some corneal endothelial cell damage after surgery is inevitable. In the literature, corneal oedema after aqueous shunt surgery due to corneal decompensation has been reported.[8] Large randomised controlled trials (RCTs) on aqueous shunt surgery such as the Ahmed vs Baerveldt Study and Ahmed Baerveldt Comparison Study had an average incidence rate of 12% persistent corneal oedema 5 years post-operation, but did not measure or report ECD loss.[9 10] The safety of glaucoma filtration surgery has progressed with the advancement from full-thickness procedures to ab interno surgery and more recently, microinvasive glaucoma surgery (MIGS). MIGS is frequently combined with cataract surgery which in itself is known to cause ECD loss.[11 12] Implantable devices used in glaucoma surgery are manufactured from different materials with varying biocompatibility profiles. The location of implants and their proximity from endothelial cells, tube stability and micromovements can impact ECD. Recently, there has been a growing use of new ophthalmic implants for treatment of glaucoma. ECD loss is an important surrogate marker for device safety and surgical failure. It is used as a requirement for some Food and Drug Administration approved investigational device exemption trials and an endpoint for postmarket approval surveillance studies. In August 2018, the MIGS device Cypass Micro-Stent (Alcon, USA) was voluntarily withdrawn from the market due to safety concerns when it was reported that 27.2% of patients had ECD loss of >30% at 5 years.[13 14] Interestingly, the device was not considered to have continuing ECD loss at 2 years in a previous study.[15]

The common cataract surgery method of phacoemulsification is where ultrasound is used to break down and aspirate the cataractous lens in the eye of a patient, followed by insertion of an intraocular lens. Cataract surgery has evolved in the last decade with new microincisional techniques which aim to cause less damage to the cornea.[16 17] This pertains to patients with glaucoma requiring cataract surgery as well.[18] Patients with glaucoma often have coexisting cataracts and will eventually require surgery for one or the other. It is possible to undergo glaucoma surgery and cataract surgery in the same operation. MIGS is frequently combined with cataract surgery. In fact, in the USA, some stents used in Schlemm's canal are only licensed for combination surgery and not for stand-alone procedures. This is reflected in trials where the intervention is combination surgery compared with phacoemulsification alone.[14 15] Combination surgery carries a distinctive risk-benefit evaluation compared with glaucoma surgery alone. Theoretically, further cost and risks of adding a short procedure to an existing operation may be less than those acquired by a separate hospital admission and anaesthetic. These combination surgeries may consequently promote a lower threshold for the use of MIGS in clinical practice. However, combination surgery can negatively affect surgical outcome in complicated cases. Since cataract surgery also causes ECD loss, it is an important confounder in many glaucoma surgery trials.

This review focuses on ECD loss from trabeculectomy, glaucoma filtration surgery and MIGS, alone or in combination with cataract surgery. A preliminary search of PROSPERO, MEDLINE, the Cochrane Database of Systematic Reviews and the *JBI Database of Systematic Reviews and Implementation Reports* was conducted and no current or underway systematic reviews on the topic were identified. There is a need to quantitatively and critically appraise available evidence and to evaluate where gaps exist.

## Review question
To determine the extent of ECD loss following different types of glaucoma surgery alone or in combination with cataract surgery in adults with ocular hypertension, primary and secondary open angle glaucoma, normal tension glaucoma and angle-closure glaucoma.

## Inclusion criteria
### Participants
This review will consider studies that include participants with ocular hypertension, primary and secondary open angle glaucoma, normal tension glaucoma and angle-closure glaucoma. There are no restrictions on geographic location, setting or demographic factors. Participants with pre-existing corneal disease such as Fuch's endothelial dystrophy, iridocorneal endothelial syndrome and corneal graft transplant and the paediatric population will be excluded.

### Intervention(s)
We will include incisional glaucoma surgery such as trabeculectomy, glaucoma drainage surgery such as Ahmed glaucoma valve implants (New World Medical, Rancho Cucamonga, California, USA), Molteno implants (Molteno Ophthalmic Limited, Dunedin, New Zealand), Baerveldt implants (Abbott Medical Optics, California, USA), and EXPRESS shunts (Alcon, USA) and MIGS (subconjunctival devices, Schlemm's canal devices, suprachoroidal devices and cyclodestruction). Glaucoma laser treatments and peripheral iridotomy will be excluded.

### Comparator(s)
We intend to compare interventions; glaucoma surgery alone or in combination with cataract surgery (phacoemulsification). The comparator could be no surgery or other type of glaucoma surgery (such as MIGS techniques or glaucoma filtration surgery) or phacoemulsification surgery.

## Outcomes
This review will consider studies that include the following outcomes:

### Primary outcome
► Percentage change of corneal ECD at 1 year.

## Secondary outcomes

► Percentage change of corneal ECD at 2 years and 5 years.
► Mean postoperative corneal ECD (cell/mm$^2$).
► Symptoms related to corneal ECD loss such as corneal oedema, bullous keratopathy or requiring subsequent keratoplasty.
► Adverse events including but not limited to intraoperative complications and hyphaema.

## Types of studies

As RCTs with ECD data is limited for glaucoma surgery, we will evaluate the studies from a broad range of studies to have a comprehensive picture and maintain generalisability without loss of validity. These include RCTs, prospective non-randomised studies and observational studies. Studies published in English language will be included. Studies published from the year 2000 to the present will be included to maintain relevance as surgical techniques have progressed and improved in the recent years.

## Methods and analysis

The proposed systematic review will be conducted in accordance with the Joanna Briggs Institute methodology for systematic reviews of effectiveness evidence.[19] Study start date June 2020 and anticipated end date June 2022.

## Search strategy

We will conduct a systematic electronic database search for RCTs, prospective non-randomised studies, observational studies in MEDLINE, EMBASE, Cochrane Central Register of Controlled Trials (CENTRAL), ClinicalTrials.gov and The International Prospective Register of Systematic Reviews (PROSPERO). The full search strategy with the keywords and index terms will be run on MEDLINE and EMBASE (see online supplemental appendix I). The reference list of all studies selected will be screened for additional studies.

## Study selection

Following the search, all identified citations will be collated and uploaded into a reference management software (Endnote X9, Clarivate Analytics) and duplicates will be removed. Two review authors will independently screen search results by title, abstract and keywords and then by full text, against the eligibility criteria, following the Preferred Reporting Items for Systematic Reviews and Meta-Analyses (PRISMA) guidelines.[20] Discrepancies between authors as to whether or not studies meet inclusion criteria will be resolved by discussion. We will document the excluded studies and reasons for exclusion, and this will be presented in a PRISMA flow diagram.[20]

## Assessment of methodological quality

Eligible studies will be critically appraised by two independent reviewers at the study level using standardised critical appraisal instruments from the Joanna Briggs Institute.[21 22] Authors of papers will be contacted to request missing or additional data for clarification, where required. Any disagreements that arise between the reviewers will be resolved through discussion, or with a third reviewer. The results of critical appraisal will be reported in narrative form and in a table.

Two review authors will assess independently the risk of bias. Critical appraisal of study methodological rigour will be performed based on critical appraisal tools,[19 23 24] depending on the experimental design of the study being assessed. Any disagreements that arise between the reviewers will be resolved through discussion. Regardless of the results of their methodological quality, will undergo data extraction and synthesis. Forest plots will also be created to graphically depict the individual and pooled effect sizes. To assess risk of bias across all studies, funnel plots for each outcome will be assessed for symmetry (using Egger test, Begg test, Harbord test where appropriate) to determine if publication bias is present.

## Data extraction

Data will be extracted from included studies by two independent reviewers aligned to the standardised data extraction tool recommended by JBI.[19 24] Variables to be extracted include:
► Study characteristics such as country of origin, year of publication and sample size.
► Trial design.
► Participant characteristics such as ages, gender, ethnicities.
► Glaucoma type.
► Surgery type and site of implant.
► ECD presurgery and postsurgery and at assessment time points.
► Number of patients withdrawn from study.
► Mean and SD for each outcome.

To minimise errors, a data extraction form has been developed for specifically this review (online supplemental appendix II). Any disagreements that arise between the reviewers will be resolved through discussion, or with a third reviewer. Authors of studies will be contacted to request additional or missing data, where required.

## Data synthesis

Studies will, where possible, be pooled with statistical meta-analysis. Effect sizes will be expressed as either ORs (for dichotomous data) or weighted (or standardised) final postintervention mean differences (for continuous data) and their 95% CIs will be calculated for analysis. Heterogeneity will be assessed statistically using the standard $\chi^2$ and $I^2$ tests. Statistical analyses will be performed using RevMan V.5.3. Subgroup analyses will be conducted where there are sufficient data to investigate ECD loss by type of surgery and glaucoma. Sensitivity analyses will be conducted to test decisions made regarding studies at high risk of bias for an outcome in one or more key domains; selection, performance, detection, attrition and reporting biases. Network meta-analysis will be conducted

where appropriate to rank type of surgery according to ECD loss. Network diagram will graphically depict the distribution of available evidence for each comparison. This review will generate comparisons between different types of surgery on ECD loss in patients with glaucoma. This will allow for comparison between surgeries that have not previously been compared. The network analysis based on a random effects model will be conservatively employed. Where statistical pooling is not possible, the findings will be presented in narrative form including tables and figures to aid in data presentation, where appropriate.

### Assessing certainty in the findings

The Grading of Recommendations, Assessment, Development and Evaluation (GRADE) approach for grading the certainty of evidence will be followed and a Summary of Findings (SoF) will be created using GRADEpro software (McMaster University, Ontario, Canada). We will grade the quality of evidence for each outcome by considering study limitations, indirectness, inconsistency, imprecision of effect estimates and risk of reporting bias. According to the software GRADEpro, we will assign four levels of quality of evidence: high, moderate, low and very low.

The SoF will present the following information where appropriate: absolute risks for the treatment and control, estimates of relative risk, and a ranking of the quality of the evidence based on the risk of bias, directness, heterogeneity, precision and risk of publication bias of the review results. The outcomes reported in the SoF will be presented in a tabular form. Outcomes for inclusion in SoF will be ECD loss.

We will present the main results of this review in a 'Summary of findings' (SoF) table, according to recommendations provided[23] in Chapter 11 of the Cochrane Handbook for Systematic Reviews of Interventions (V.5.1.0). We will provide estimates derived from the meta-analysis in accordance with methods of the GRADE (Grades of Recommendation, Assessment, Development and Evaluation) Working Group.[25]

### Ethics and dissemination

Ethics approval is not required for this review, as it will only include published data. Findings will be published in a peer-reviewed journal and disseminated across ophthalmic networks. We anticipate that the findings of this work will be of interest to multiple stakeholders: people who have undergone glaucoma surgery, eye health professionals, ophthalmic surgeons, device manufacturers and policy makers. It will also inform researchers to where there are gaps in evidence and identify areas for future research.

**Contributors** CH conceived the idea for the review. CEHF and CH drafted and revised the protocol with suggestions from PTK and RGM who reviewed the protocol and provided feedback on the draft. CEHF constructed the search.

**Funding** The authors receive funding from the National Institute for Health Research Biomedical Research Centre at Moorfields Eye Hospital NHS Foundation Trust and UCL Institute of Ophthalmology, the UK Medical Research Council, Moorfields Eye Charity, the Michael and Ilse Katz Foundation, the Helen Hamlyn Trust and Fight for Sight (UK).

**Competing interests** None declared.

**Patient consent for publication** Not required.

**Provenance and peer review** Not commissioned; externally peer reviewed.

**Data availability statement** Data sharing not applicable as no datasets generated and/or analysed for this study. All data relevant to the study are included in the article or uploaded as supplementary information.

**ORCID iD**
Christin Henein http://orcid.org/0000-0002-6972-5355

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
