## [Reviewer comments · BMJ Open]

ARTICLE DETAILS

TITLE (PROVISIONAL)	Corneal endothelial cell density loss following glaucoma surgery alone or in combination with cataract surgery: A systematic review protocol
AUTHORS	Fang, Clarissa E H; Khaw, Peng; Mathew, Rashmi G; Henein, Christin

VERSION 1 – REVIEW

REVIEWER	Hlrooka, KAzuyuki Hiroshima University
REVIEW RETURNED	07-May-2021

GENERAL COMMENTS	To investigate corneal ECD loss following glaucoma surgery is very important. Therefore, systematic review of this topic is necessary. A decrease in corneal ECD is a serious complication after glaucoma drainage device surgery. Glaucoma drainage device was inserted into pars plana, anterior chamber, or sulcus placement. Probably the rate of corneal ECD loss depends on where the tube was inserted. Therefore, the authors should determine the extent ECD loss following not only different type of glaucoma surgery but also different placement insertion of the tube.
--

REVIEWER	Kitazawa, Koji Kyoto Prefectural University of Medicine, Department of Ophthalmology and Department of Frontier Medical Science and Technology for Ophthalmology
REVIEW RETURNED	17-Jul-2021

GENERAL COMMENTS	Endothelial loss after glaucoma surgery is an important clinical question. A systematic review is meaningful because there is limited RCT studies with ECD loss post glaucoma surgery. Doesn't the glaucoma subtype need to be considered? What about iris damage? There are studies that are looking at the association between ECD and iris damages. cf. Sci Adv. 2020 May 13;6(20):eaaz519, Sci Rep. 2016 Apr 28;6:25276. Is it considered glaucoma eye drops pre and post surgery? The presence of ROCK inhibitor could influence the postoperative ECD.
--

VERSION 1 – AUTHOR RESPONSE

Reviewer 1:

1. Where possible the extent of ECD loss following different placements of the device will be determined (e.g. pars plana, anterior chamber, or sulcus placement). Line ..

Reviewer 2:

1. We will more clearly specify glaucoma subtype as one of the data extraction variables.
2. Glaucoma eye drops and duration of use are valid confounding variables. Studies poorly report these variables, however we will endeavour to report them where possible.
3. Iris damage/touch and certainly intra-operative endothelial damage can ensue, where studies have reported this we will endeavour to summate. Thank you for providing references.